# Cognitive causes of 'like me' race and gender biases in human language production

Jessica Brough [1] ✉, Lasana T. Harris [2], Shi Hui Wu [1], Holly P. Branigan [1] & Hugh Rabagliati [1]

Natural language contains and communicates social biases, often reflecting attitudes, prejudices and stereotypes. Here we provide evidence for a novel psychological pathway for the expression of such biases, in which they arise as a consequence of the automatized mechanisms by which humans retrieve words to produce sentences. Four experiments show that, when describing events, speakers tend to mention people who are more like them first and, thus, tend to highlight the perspectives of their own social groups. This 'like me' effect was seen in speakers from multiple demographic groups, in both English and Chinese speakers and in both first- and second-language English speakers. Psycholinguistic manipulations pinpoint that the bias is caused by greater accessibility in memory of words that refer to in-group than out-group members. These data provide a new cognitive explanation for why people produce biased language and highlight how detailed cognitive theories can have social implications.

Natural language contains and communicates a variety of social biases[1–6], and this fact increasingly impacts human life, from education[7–9] to corporate communication[10–12] and the development of artificial intelligence[13,14]. Large language models, for example, when naively trained on text data harvested online, demonstrate a variety of unintended negative traits such as stereotypical or derogatory associations with social features such as gender, race and disability[4,5,14]. The causes of social biases within human language are diverse, including not just the beliefs, prejudices and goals of individuals and social groups[6,11,15–18], but also historically ingrained patterns of use for particular words and phrases[8,19,20]. Here, however, we provide evidence for a previously undocumented psychological cause of biased language, in which it arises as a consequence of the automatized mechanisms by which humans retrieve words from memory to produce sentences.

A key design feature of human sentence production is its incrementality: rather than planning long sequences of words before saying them, speakers instead retrieve words from memory in a just-in-time fashion, so that the first words to be retrieved are also the first words to be spoken, thereby promoting fluency[21–25]. The order by which words are retrieved is generally guided by an approximate plan of the message that the speaker intends to communicate, such as who did what to whom[26–28]. However, it

can also be the case that the creation of these messages is influenced by actual or potential difficulties and delays in accessing the specific words that the speaker needs to produce[27,29]. For example, an English speaker describing two animals playing might typically describe it as 'the dog is chasing the cat', a frame that highlights the dog as the agent of the event by placing its label as the first-mentioned sentence subject. However, if the speaker only retrieves the word for the agent from memory after the word for the patient (for example, if the speaker's attention had been initially drawn to the cat before they noticed the dog), then this can cause the speaker to produce a description that places the first retrieved word first and, thus, changes both the communicated perspective on the event and the message: 'the cat is running away from the dog'[27].

Changes in word order and perspective like this may seem subtle, but they have important consequences for how listeners and readers understand language and the world. First-mentioned words are processed more deeply[30], serve as starting points that frame subsequent discourse[31] and are considered more relevant and central for the event being described[32]. Moreover, in active-voice transitive sentences where the first-mentioned word is the sentence subject (for example, 'the man is meeting the woman'), that first-mentioned individual will be considered more agentic than the second-mentioned individual, that is, more

[1]School of Philosophy, Psychology and Language Sciences, University of Edinburgh, Edinburgh, UK. [2]Department of Experimental Psychology, University College London, London, UK. ✉e-mail: jessicabrough3@gmail.com

responsible for the event occurring[33] and more active and potent[34], while their perceived social identity will also be more closely bound to that of the person producing the sentence[35]. Thus, because accessibility and incrementality influence how people describe events, they can have a knock-on effect in terms of how those events are framed for other people. A social bias in incremental production could therefore strongly impact how listeners construe the importance and agency of different social groups.

The accessibility of a word, and its impact on ordering in production, is known to be influenced by both language-internal and cognitive factors, like the structure of discourse so far[36], the organization of the mental lexicon and the structure of semantic memory[37,38]. For instance, words for concepts that are more prototypical[37,39], more imageable[40] or more animate[41] are also typically easier to access and, thus, more likely to be mentioned first, which in turn means that speakers give these concepts greater highlighting. However, alongside these basic linguistic and cognitive determinants, there are good reasons to believe that further social properties of a word and its referents could also impact the process of incremental access, and that these social properties may inadvertently cause people to produce biased language. For instance, recognition of out-group members, such as people of different genders or races, is slower and less accurate than recognition of in-group members[42–46]. A knock-on effect of this may be that access to the names of any out-group members will be delayed relative to the names of any in-group members. Given that the production system is incremental, operating in the first-in-first-out fashion described above, the consequence would be a bias to generate descriptions that start with the names of in-group members rather than out-group members and, thus, inadvertently highlight the in-group perspective over the perspectives of any out-groups. We term this potential bias the 'like me' effect, building on prior claims about historical and conventionalized social biases in word order[20]. Some possible evidence for its occurrence comes from the production of binomial expressions, like 'Sally and John': when naming pairs of friends, people are more likely to start with (and thus highlight) the person who is socially closer to them[47], perhaps because the more familiar name is also more accessible.

Importantly, this processing-based account of biased language is distinct from more traditional social psychological, sociolinguistic and grammatical accounts[2,3,16,48]. These typically construe social biases in language as a consequence of the expression of social agency, status and affiliation. This can emerge as a tendency to highlight the agency of a speaker's own group[16,49], to emphasize a speaker's identification with their group[35] or to foreground the importance of socially dominant figures or groups, as in the case of androcentric language (for example, 'his and hers, king and queen')[7,15,20]. On our account, by contrast, linguistic biases do not only result from historical conventions, or the messages that speakers implicitly choose to convey, but can also be a consequence of the particular cognitive mechanisms that we use to efficiently choose and order our words.

We therefore investigated how the social categories of gender and race influence ordering in language production. In our first three studies (Fig. 1a,b), participants learned the names of eight people, half female and half male, half Black and half white, and then described events in which two of the people, differing in either gender or race, interacted in a symmetrical fashion, such as meeting each other, dancing with each other or talking with each other[50]. Under these conditions, the same event could be described from either person's perspective but always using the same verb and grammatical frame, so that the description 'Beth is dancing with Ruth' communicates the same denotational meaning as the description 'Ruth is dancing with Beth', yet highlights different connotations of agency based on the perspective that was chosen. We then measured the degree to which participants framed their descriptions in ways that highlighted their own social group, by beginning their sentences with the name of the person who matched them in gender or in race.

In experiment 1, 240 online participants (60 Black women, 60 Black men, 60 white women and 60 white men, recruited from English-speaking countries available through the website Prolific) described events similar to those in Fig. 1a. At the start of the task, participants were shown eight figures that they would be describing (two Black women, two Black men, two white women and two white men) and were told to study their names, each of which was frequent and monosyllabic but not gender neutral, for example, 'Luke, Ruth'. Names were also counterbalanced between the Black and white figures for half of the participants within each social group. No further social information about the figures was provided beyond their photograph and name. After learning each character's name to ceiling accuracy (tested in an eight-alternative forced choice task that additionally repeated every 20 trials throughout the study), participants then provided descriptions of 80 events based on 10 symmetrical verbs. Half the events showed a woman interacting with a man (matched for race), and half showed a Black person interacting with a white person (matched for gender). On each trial (Fig. 1b), participants were first shown the event as a photograph that mirror-reversed every half second for 2 s so as to minimize participants' tendency to begin their descriptions with the figure shown on the photograph's left-hand side. Next, participants were shown a sentence frame for 2 s that they would use in their description. The frames were either transitive ('… is meeting …') or intransitive with a conjoined subject ('… and … are meeting each other'). This manipulation of sentence structure was designed to separate our incrementality-based account from more traditional social psychological and sociolinguistic accounts. Specifically, the subjects of transitive sentences are typically interpreted as being more agentive than non-subjects[33]. So, if speakers act to highlight the agency of their in-group, or communicate their identification with that in-group, then we should expect that this effect would be greater in the transitive condition, where only one name can be a subject, than in the intransitive condition, in which both names are grammatical subjects and therefore agents of the event, and where the phrase 'each other' should create greater symmetry between their roles[50]. However, if accessibility simply causes participants to name the in-group member first, as our account predicts, then we should find a similar effect for both frames. Thus, having seen the image and read the sentence frame, participants finally typed out a full description, for example, 'Ruth and Luke are meeting each other'.

We applied pre-registered exclusion criteria to these data (for example, to remove participants who failed to engage with the task; Methods) and then analysed whether participants showed a like me effect, measuring the percentage of trials on which they began their sentence by naming the person who either matched them on gender (when the figures differed in gender but not race) or matching them on race (when the figures differed in race but not gender). Participants showed a robust tendency to mention the person like themselves first, doing so on 54.9% of trials (bootstrapped 95% confidence interval (CI) 53.6–56.2, $\beta = 0.22$ (s.e.m. 0.03), $z = 7.3$, $P < 0.001$; statistical analyses conducted using mixed-effects logistic regressions). This effect was present in each of the four participant groups when analysed separately (Black women: 52.3% (50.1–54.9), $\beta = 0.12$ (0.06), $z = 2.1$, $P = 0.036$, Black men: 56.6% (53.8–59.8), $\beta = 0.28$ (0.07), $z = 3.9$, $P < 0.001$, white women: 53.6% (51.1–56.2), $\beta = 0.15$ (0.06), $z = 2.6$, $P = 0.010$, white men: 57.3% (54.8–59.8), $\beta = 0.33$ (0.06), $z = 5.7$, $P < 0.001$) and was also present when we separated the gender trials (55.6% (53.6–57.9), $\beta = 0.29$ (0.05), $z = 5.4$, $P < 0.001$) and the race trials (54.2% (52.7–55.6), $\beta = 0.19$ (0.04), $z = 5.5$, $P < 0.001$). The analysis of the manipulation of sentence syntax was more equivocal, however. Overall, the like me effect was not significantly increased when participants used a transitive sentence frame (56.6% (54.4–58.7)) over a conjoined subject intransitive sentence frame (53.3% (50.9–55.6), $\beta = 0.07$ (0.04), $z = 1.7$, $P = 0.084$). However, when gender trials were analysed alone, we found that the effect was larger for transitive than intransitive frames ($\beta = 0.14$ (0.07), $z = 2.1$, $P = 0.040$),

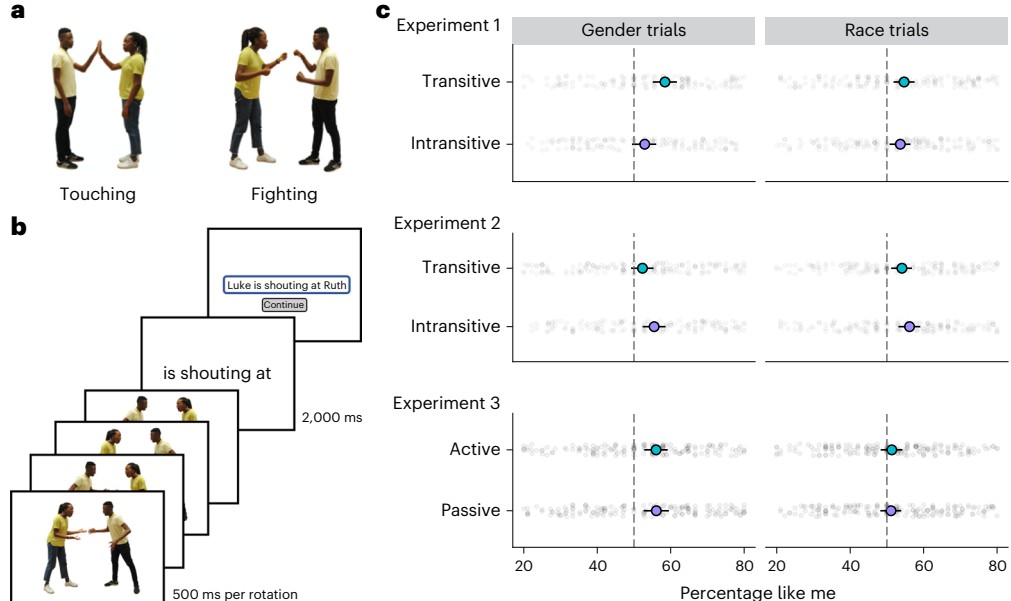

**Fig. 1 | Evidence for a like me effect in language production. a**, Example events (from stimuli used in experiment 2). **b**, Instructions for the trial procedure. Participants saw a reversing image followed by a sentence frame, and then typed a response using the figure names and the frame. They completed 40 gender trials and 40 race trials. **c**, Results from experiments 1 through 3. Percentage of trials on which participants' descriptions began with the person like them, on gender trials and on race trials, split by grammatical frame (experiment 1, $n$ = 196; experiment 2, $n$ = 195; experiment 3, $n$ = 196). The large dots show conditional means with bootstrapped 95% CIs, and the small dots show individual participant means. The dashed black lines mark unbiased (50%) behaviour. The axes are trimmed from 20% to 80% to highlight the critical data region.

although for race trials there was no significant evidence for a difference based on syntax ($\beta$ = 0.03 (0.06), $z$ = 0.4, $P$ = 0.676).

The key finding of experiment 1 is the existence of a previously undocumented like me effect in language production: speakers highlight members of their social in-groups by probabilistically placing expressions referring to those individuals at the start of a sentence. We found this both for sentence subjects and for the first-mentioned item in a conjoined subject. We suggest that this like me effect is driven by the names of in-group members being more accessible and therefore produced first, following first-in-first-out incrementality. In turn, we do not suggest that the effect occurred because speakers implicitly designed their utterances to highlight their in-group, either in terms of communicating its agency or their identification as a group member. That latter account predicts that we should find consistent effects of grammar on production, with speakers being more likely to put their in-group first when using transitive frames (where the in-group member is more clearly highlighted) than intransitive frames (where the roles are shared). However, this prediction was not met: overall, grammar had no significant influence on the size of the like me effect. That said, subjects were more likely to show a like me effect for gender when producing transitive rather than intransitive sentences, although this did not hold for race, meaning that this first study did not entirely settle the issue.

Critically, we emphasize that effects of accessibility on linguistic biases were not simply driven by the forms of the names being more familiar within a social group. In this task, the names of the Black and white figures were matched and counterbalanced, and yet race still caused a like me effect. Thus, this effect is not driven by some names being easier to recall from memory than other names overall, but rather by dynamic differences in the accessibility of a name that depend upon the particular social identity match between the person speaking and the person being named.

In experiment 2, we aimed to confirm the results of experiment 1 by replicating it with a new set of stimuli that minimized non-critical visual differences between the displayed figures. A total of 240 new participants (again, 60 Black women, 60 Black men, 60 white women and 60 white men recruited through Prolific) followed a protocol that was otherwise matched to experiment 1, including the manipulation of sentence structure (active transitive versus conjoined subject intransitive).

Experiment 2 replicated the key like me finding of experiment 1. Participants were significantly more likely to begin their descriptions with the name of the person who matched them in either gender or race (mean 54.4% of trials, 95% CI 53–55.6, $\beta$ = 0.19 (0.03), $z$ = 6.4, $P$ < 0.001). This effect was statistically significant for all social groups except Black women (Black women: 51.9% (49.1–55.4), $\beta$ = 0.09 (0.06), $z$ = 1.5, $P$ = 0.128, Black men: 57.5% (54.8–59.9), $\beta$ = 0.31 (0.06), $z$ = 5.1, $P$ < 0.001, white women: 54% (52.1–56), $\beta$ = 0.16 (0.04), $z$ = 3.6, $P$ < 0.001, white men: 54.8% (52–57.6), $\beta$ = 0.23 (0.07), $z$ = 3.1, $P$ = 0.002) and also held when the gender (53.9% (51.9–55.7), $\beta$ = 0.18 (0.04), $z$ = 4.2, $P$ < 0.001) and race trials (55% (53.3–56.8), $\beta$ = 0.22 (0.04), $z$ = 5.5, $P$ < 0.001) were analysed separately. Importantly, unlike in experiment 1, we found no evidence—for either gender or race trials—that the syntactic frame affected the size of the like me effect. Indeed, in this study participants showed a numerically stronger effect on conjoined subject trials rather than transitive trials (conjoined: 55.7% (53.5–58.1), transitive: 53.1% (50.8–55.4), $\beta$ = −0.06 (0.04), $z$ = 1.3, $P$ = 0.183). This null finding lends additional weight to our proposal that the like me effect is driven by accessibility, rather than a tendency to highlight the agency of the in-group.

However, in the studies so far, participants were always required to produce sentences in which the first-mentioned individual was also an agent of the event, meaning that those data cannot fully rule out that speakers are acting to highlight agency. Experiment 3 provided evidence against this alternative interpretation and in favour of the accessibility-driven explanation of the like me effect. A total of 240 American participants from the same four social groups described interactions, but this time we instructed them to use either active sentences ('…is hugging…') or passive sentences ('…is being hugged by…'), in which the first-mentioned sentence subject is now the patient,

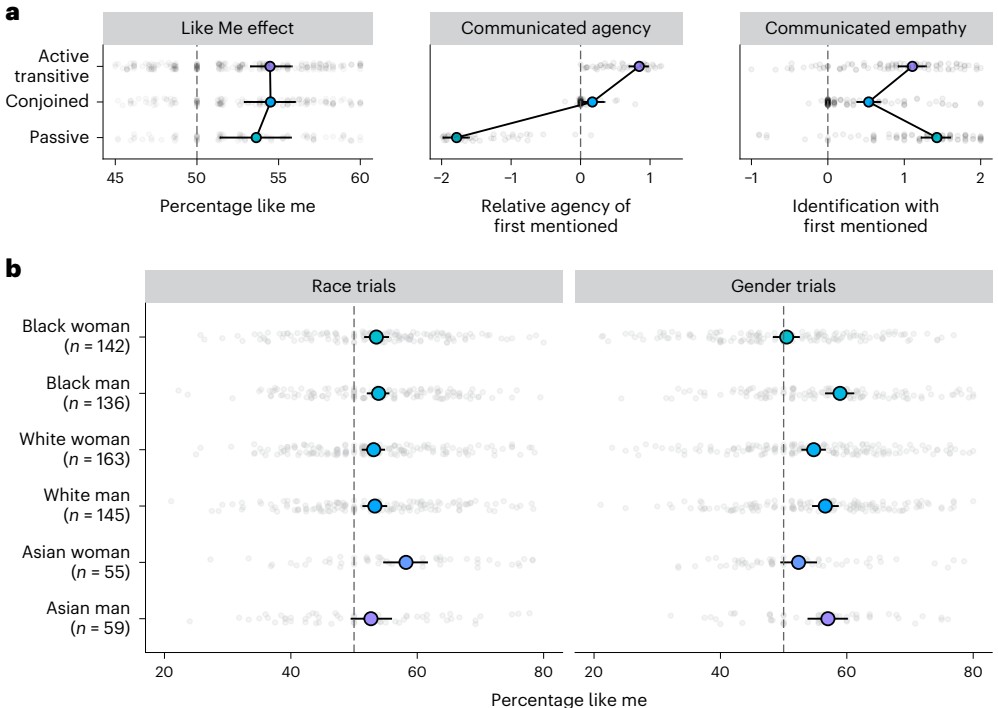

**Fig. 2 | The like me effect dissociates from communicated empathy and agency, and holds across groups. a**, Mean percentage of trials on which participants' (*n* = 587) descriptions began with the person like them in experiments 1 through 3, split by grammatical constructions, alongside ratings of communicated agency of the first-mentioned person (from *n* = 171 participants), and ratings of a speaker's identification with the first-mentioned person (from *n* = 109 participants), also split by construction. **b**, The size of the like me race and gender effects across the six demographic groups combining experiments 1 through 4 (the large dots show conditional means with bootstrapped 95% CIs, the small dots show individual participant means, the sample sizes are reported directly in the figure, and note that the Asian sample size is smaller and estimated effects are noisier).

not the agent, of the event, and the agent is only mentioned second as the sentence's object. If the like me effect arises from an implicit attempt to highlight the agency of the in-group, we would expect it to be reversed in the passive-voice condition, so that participants now tend to mention the in-group member second as the object of the sentence. The methods here were very similar to those of experiment 2, except that we had to replace seven of the ten events with new items where the verbs could be made passive.

As before, we found robust overall evidence for a like me effect (*M* = 53.6% of trials (52.4–54.8), $\beta$ = 0.15 (0.03), *z* = 5.6, *P* < 0.001). The effect was directionally consistent but not significant in Black women, and significant in the other social groups (Black women: 51.9% (49.3–54.5), $\beta$ = 0.10 (0.06), *z* = 1.6, *P* = 0.106, Black men: 55.4% (52.6–58.3), $\beta$ = 0.22 (0.06), *z* = 3.7, *P* < 0.001, white women: 54.2% (52.2–56.2), $\beta$ = 0.17 (0.04), *z* = 3.8, *P* < 0.001, white men: 52.5% (50–54.8), $\beta$ = 0.12 (0.05), *z* = 2.2, *P* = 0.026). The like me effect was statistically significant when the gender trials were analysed alone (55.9% (53.9–57.7), $\beta$ = 0.28 (0.05), *z* = 5.2, *P* < 0.001), and directionally consistent but not significant when the race trials were analysed alone (51.2% (49.6–52.9), $\beta$ = 0.06 (0.04), z = 1.6, *P* = 0.096). Critically, the manipulation of active and passive voice had no impact on the size of the effect: There was no evidence that participants showed a reduced like me effect when producing passive sentences (53.6% (51.7–55.7)) as compared with active sentences (53.7% (51.6–55.8), $\beta$ = 0.0000967 (0.039), *z* = 0.002, *P* = 0.998). Thus, these data indicate that participants tend to mention their in-group first and do not act to highlight their in-group's agency, consistent with an explanation of the like me effect that is based on accessibility.

Our data so far demonstrate a like me effect that is invariant across the grammatical constructions that participants were required to produce. This held despite there being strong intuitive differences in the information that these constructions communicate about factors like

agency and group identity. To confirm these intuitions, we asked a new set of 192 British participants (half female) to rate the communicated agency of the two mentioned names in transitive active sentences like 'Ruth is hugging Luke', conjoined subject sentences like 'Ruth and Luke are hugging each other' and passive-voice sentences like 'Ruth is being hugged by Luke'. We also asked a further 128 British participants to rate whether a person speaking these sentences would be perceived to identify more with the first- or second-mentioned name, following claims that grammatical constructions act to highlight the individual with whom the speaker identifies[35]. We then compared these ratings with the like me effects found in experiments 1 through 3. As Fig. 2a shows, the sizes of the like me effects varied only minimally across the grammatical constructions that we had tested ($\chi^2(2)$ = 0.65, *P* = 0.721). Rated agency and rated identification, by contrast, strongly varied based upon the grammatical constructions (both $\chi^2(2)$ > 200, *P* < 0.001) and clearly dissociated from one another. The fact that the like me effect is invariant rather than similarly variable suggests that it is instead driven by a constant factor, such as accessibility.

To confirm the impact of demographic match on accessibility, we turned to the name learning task that had been embedded in our three studies. Throughout experiments 1 through 3, participants were repeatedly tested on the names of each figure using an eight alternative forced choice task; we therefore pre-registered an analysis of the time to make these responses. A pre-registered mixed-effects regression found that response times significantly increased as demographic match decreased ($\beta$ = 0.02 (0.005), *t*(642) = 4.5, *P* < 0.001): participants were fastest to select names when they matched the figures on both gender and race (*M* = 2,724 ms (2,642–2,801)). When participants matched the figures in gender but differed in race (and, thus, the forms of names were perfectly counterbalanced), then responses were about 90 ms slower (*M* = 2,813 ms (2,734–2,895)), and when participants matched in race but differed in gender, then responses were about 70 ms slower

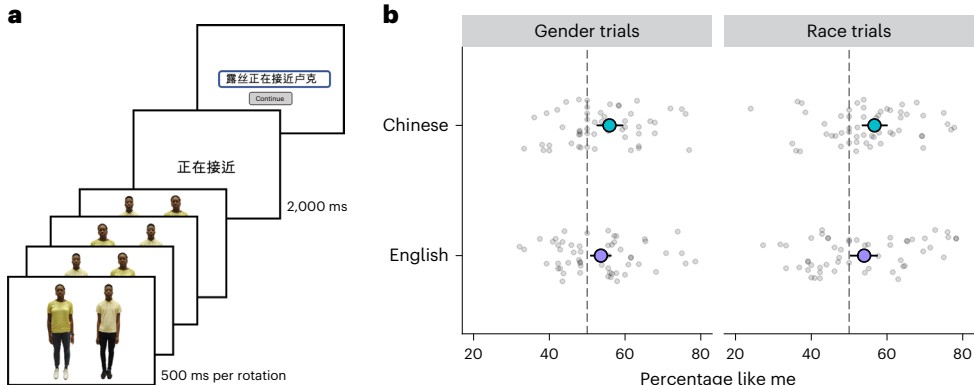

**Fig. 3 | The like me effect in Chinese bilinguals. a**, Instructions for the trial procedure. Participants saw a reversing image followed by a sentence frame and then typed a response using the figure names and the frame. **b**, The percentage of trials on which participants' descriptions began with the person like them, on gender trials and on race trials, split by the language used (first language Chinese or additional language English).

($M$ = 2,794 ms (2,705–2,881)). Finally, when participants differed from the figures in both gender and race, they were an additional 50 ms slower to name them ($M$ = 2,856 ms (2,771–2,943)). Thus, participants were faster to report the names of people who were more like them, indicating that those names are indeed more accessible in memory. This accessibility advantage naturally explains the like me effect.

These experiments reveal a clear social linguistic bias in white and Black native speakers of English. How generalizable is it? In experiment 4, we tested whether a similar like me effect is found in East Asian native speakers of Chinese, both when using their first language and when using English as an additional language. We used a slightly modified experimental design in which participants no longer described two figures interacting in an event but instead were shown a mirror-reversing pair of figures standing side by side in a non-interacting fashion. Then, participants were given a sentence frame and finally they produced a sentence using the figures' names and the frame (Fig. 3a). This design (which pilot work showed elicited a like me effect in English) therefore differed from experiments 1 through 3 in that we did not depict events directly but rather allowed participants to imagine them. This meant that we could now use a distinct set of non-symmetric verbs as stimuli and, thus, further assess generalization across linguistic materials. Participants were 120 Chinese students, native speakers of standard Chinese (Mandarin) who also had learned English as a second language and now studied at English-speaking universities (60 female and 60 male). Half completed the study in standard Chinese and half in English, using ten translation-equivalent English and Chinese verbs. We also used Chinese translations of the English names (a side effect of which was that two of the female names did not have a strong gender bias). In this study, there was no manipulation of syntax and participants only produced transitive active sentences. The white figures from the previous experiments' stimuli were replaced with two East Asian women and two East Asian men, so that trials focusing on race now compared Black and East Asian figures matched for gender, while trials focusing on gender compared male and female figures matched on race. At the start of the study, participants learned the names for the figures in the same way as before, and again no background information was given about the figures beyond their names and photographs.

Under these conditions, the like me effect generalized robustly (Fig. 3b; $M$ = 55% of trials (53.4–56.8), $\beta$ = 0.18 (0.04), $z$ = 5.0, $P$ < 0.001). It was present when participants natively produced Chinese (56.2% (53.8–59.0), $\beta$ = 0.23 (0.06), $z$ = 4.1, $P$ < 0.001) and also when they produced English as an additional language (53.7% (51.7–55.6), $\beta$ = 0.13 (0.05), $z$ = 3.0, $P$ = 0.002). The size of the effect did not significantly differ depending on the language used ($\beta$ = −0.05 (0.04), $z$ = 1.3, $P$ = 0.192).

The effect was present in both men (54.7% (52.3–57.3), $P$ < 0.001) and women (55.3% (53.2–57.4), $P$ < 0.001). Thus, the like me effect is present across two typologically different languages, English and Chinese, and appears to hold no matter whether a person is speaking their native language or an additional language.

These experiments reveal a social bias that not only holds consistently across different linguistic constructions and different languages, but that also holds similarly across different social groups. It was not simply the case that participants were biased to mention socially dominant figures first, or that only socially dominant groups, like white men, showed a bias to mention in-group members first—all of our measured groups showed a like me effect. But was the size of the like me effect similar across groups, or were some groups more likely to take their own perspective? We estimated this by combining the individual-level data across experiments 1 through 4. We analysed whether the overall size of the Like Me effect was predicted by participant gender, participant race and their interaction.

We first did this for the race trials alone: because names were perfectly counterbalanced in this condition, it provided a measure of the like me effect that was not impacted by other confounds such as lexical frequency. As Fig. 2b shows, the like me effect was present across groups ($\beta$ = 0.16 (0.02), $z$ = 7.7, $P$ < 0.001), but an interaction between gender and race indicated that the like me effect for race was slightly larger in Asian women ($\beta$ = 0.27 (0.11), $z$ = 2.4, $P$ = 0.0183), although note that estimates for this group were noisy because the sample size was small. Next, we compared the like me gender effect across groups. Importantly, because we used distinct male and female names, this comparison could be influenced by specific lexical characteristics, such as androcentric biases that make male names more globally frequent or prominent[51]. Again, we found that a like me effect held overall ($\beta$ = 0.21 (0.02), $z$ = 8.5, $P$ < 0.001), but it was significantly smaller in women than in men ($\beta$ = −0.2 (0.05), $z$ = 4.2, $P$ < 0.001). Moreover, and importantly, an interaction between gender and race indicated that the size of the like me gender effect was even smaller in Black women ($\beta$ = −0.28 (0.09), $z$ = 3.0, $P$ = 0.003). Indeed, this group did not actually show a significant like me gender effect ($M$ = 50.5 (48.3–52.7), $\beta$ = 0.02 (0.05), $z$ = 0.49, $P$ = 0.623). The causes of these differences, and the null like me gender effect for Black women, clearly warrant further examination. For example, it could be a consequence of Black women having specific and distinct experiences with androcentric language, or it may reflect differences in how social groups experience and demonstrate subjectivity. Finally, we also directly compared the size of the like me effect for race and gender across all four studies, showing that the effect for race was overall slightly smaller ($M$ = 53.7% (52.8–54.6)) than the effect for gender ($M$ = 55.1% (54.1–56.1), $\beta$ = 0.04 (0.02), $z$ = 2.1, $P$ = 0.0356).

The experiments here provide clear converging evidence for a novel cause of linguistic biases, the like me effect, in which speakers probabilistically begin their utterances with the names of people who match their own demographic characteristics. The like me effect occurred whether the demographic match was in terms of race or gender, and it was present across a variety of conditions and populations: in both women and men, in Black, Asian and white participants, for speakers of both English and Chinese, for both first and additional languages, and across a range of grammatical frames. The fact that grammatical frame, in particular, did not affect the size of the like me effect provides evidence as to its origin, specifically the process of incrementally accessing words during language production, a conclusion reinforced by the finding that participants were also faster to retrieve the names of characters who matched their own demographics, even when the forms of those names were precisely matched.

Decades of detailed psycholinguistic research have characterized the influence of accessibility on language production, highlighting how incrementality causes speakers to begin their utterances with words that are more accessible and showing how accessibility is itself determined by factors such as lexical frequency, semantic richness and discourse structure[36–38,52,53]. The present results, however, show striking evidence for a hitherto underexplored determinant of psycholinguistic accessibility, in terms of social identity. A number of factors could potentially cause the names of in-group members to become more accessible. For example, before accessing a person's name, the speaker must first recognize their face, and this recognition process is known to be delayed and less accurate for out-group members, whether defined in terms of race[46], gender[44] or even age[42] and affiliation[43]. Importantly, under this pathway, the accessibility of a name would not be determined by difficulty searching through the lexicon (unlike more standard considerations of accessibility),¡ but rather by socially defined perceptual and attentional processes that could influence how speakers view and construe the events that they are talking about. The knock-on effects of these processes would be to predictably stagger the order in which words are accessed for language use, resulting in linguistic biases.

These data highlight how social identity can influence language use in ways that go beyond standard social psychological and sociolinguistic accounts[2,3,49]. Social psychological theories have typically focused upon speaker's tendencies to highlight the agency of, or their own identification with, socially dominant groups and in-groups[15,16,49]. For example, research on androcentrism reports biases to put men first across a variety of tasks, whether writing names or drawing figures[54–57]. Our data should not be taken to refute these claims: We found evidence for androcentrism in how the like me effect for gender was smaller in women than men. However, androcentrism was clearly not the only contributor to participants' behaviour, as indicated by the like me effect overall being present in both women and men, and also by the key fact that, where we could experimentally control for lower-level properties of the names used, in the race conditions, we did not see any evidence that socially dominant groups were given special status. For example, participants were not overall biased to begin their descriptions with the names of white figures. Our data also indicate that the like me effect is not a consequence of speakers acting to highlight or promote their in-group. We found that participants were just as likely to begin their sentences with their in-group member when assigning them to be the agent of an event (in an active transitive sentence) or to be acted upon (in a passive-voice sentence), as well as when using a grammatical frame (coordinated subject intransitives) that served to minimize the speaker's communicated empathy for that first-mentioned person. The fact that the like me effect was invariant across constructions suggests that it is driven by accessibility and incrementality, rather than by evaluation.

Our evidence indicates that the like me effect holds across a variety of contexts and populations, including native speakers of English, native speakers of Chinese and speakers of Chinese using English as an additional language. This suggests that the same biases and effects are likely to be found beyond the particular populations and the specific languages tested here. However, one potential concern with this claim is that—in certain respects—the two languages that we tested are actually grammatically somewhat similar. While English and Chinese are only very distantly related typologically, they both use the same strategy of marking grammatical roles through a relatively fixed subject–verb–object word order, whereas other languages may use either different fixed word order patterns (such as subject–object–verb in Turkish) or use less-fixed orders and mark grammatical roles through morphology. It thus remains strictly open how the like me effect generalizes to these different language types. Nonetheless, we predict that the effect would still be present, as psycholinguistic experiments on a range of languages consistently find that accessibility and incrementality play an important role in production no matter the apparatus used to mark grammatical roles[58–60].

Our study finds evidence for a like me effect based on two perceptually salient dimensions of social identity, and we would also expect to find it in similarly salient dimensions such as age. Left open, however, is whether like me effects are also found for dimensions that are socially important but less perceptually salient, such as political associations or religion. Moreover, it is also still unclear how this effect may be modulated when social identity is more graded or intersectional. One possibility is that a more nuanced conceptualization of identity may explain the somewhat different patterns that we observed for gender and race in these studies, such as a slightly smaller overall like me effect for race than for gender, and a consistently absent like me effect for gender in Black women. To wit, the former result may derive from race being experienced and construed in an even more multi-dimensional fashion than gender, incorporating factors such as skin tone, national identity and a sense of identifying as mixed race rather than as within any particular category. The latter result may be explained along similar lines, perhaps in terms of how gender and race intersect in complex and unique ways for all social groups. This uniqueness for Black women derives from the combination of historical stereotypes, prejudices, pressures and resistances[61], which may in turn affect subtle linguistic choices of mention order and perspective-taking.

Most importantly, these psycholinguistic results have real-world implications because they reveal clearly recognizable psychological determinants of why people might use biased language. The measured like me effect across these studies is in some sense small, with mean values at approximately 55%. However, given the importance of language use in daily life, the effects of even a relatively limited bias will still be widespread. For individual speakers, the like me effect may cause them, even if unintentionally, to communicate positive associations with people from their in-groups and, potentially, negative associations with their out-groups. More broadly, the like me effect offers a novel mechanistic explanation of why certain patterns and features of language have evolved. These include conventionalized androcentric orderings, such as 'his and hers' or 'man and woman', but also cross-linguistic grammatical generalizations, such as the grammatical tendency to use sentence subjects to highlight the speaker's perspective, and the placement of those subjects early in the sentence[20,35,51,53]. While these conventionalized patterns may not be proximally caused by like me accessibility today, it may have steered them over historical time, guiding the process of conventionalization[62]. In this way, the like me effect may serve as an upstream cause for historically ingrained patterns and structures.

Finally, a core conclusion from the present studies is that the incremental and accessibility-driven language system that allows speakers to produce language that is fluent also leads them to produce language that is biased. However, we do not intend to imply that the unintentional nature of this effect means that we should simply accept biased language as a cognitive inevitability. Instead, these data can serve as a starting point for strategies that can act to minimize linguistic biases, a particularly important problem in the development of artificial

intelligence and language models. They highlight that, when particular voices are privileged, the perspectives and agency of those groups will also be automatically highlighted for reasons of basic cognitive psychology. Ensuring the deliberate and equitable representation of voices across groups can therefore provide, to some degree, an automatic and implicit remedy for inequalities of perspective-taking and agency.

## Methods
This research complied with all ethical regulations of and was approved by the Psychology Research Ethics Committee at the University of Edinburgh number 248-1819/3. Informed consent was obtained from all participants.

### Experiment 1
The pre-registration for experiment 1 can be found at https://osf.io/bfhrn. We recruited 240 adults from Prolific.com (60 Black women, 60 Black men, 60 white women and 60 white men). The sample size was set without a power analysis, based on the maximum number of participants we could reasonably recruit with the study budget. A subsequent power analysis using the procedures described in Westfall, Kenny and Judd[63] and the online application jakewestfall.shinyapps.io suggests that this sample size combined with our number of trials (80 per participant) gives 80% power to detect an effect size of 0.23. For subgroup analyses of 60 participants, we had 80% power to detect effect sizes of 0.31.

Participants were all 18 years old or above, self-defined their ethnicity as either Black or white, their sex as either male or female, their gender identity as either male or female and their first language as English, and had reported no language-related disorders. They were compensated £4 for an approximately 30 min study.

Participants first completed a memory task to learn the names of the eight figures they would be describing, and then completed the picture description task. The memory task began by presenting an array of the eight figures and their names, which participants were told to memorize. Then, participants were asked to recall the name of each figure individually using an eight alternative forced choice task. If participants did not reach 100% accuracy, the task repeated until they did. The same memory task then repeated every 20 trials to ensure participants maintained naming accuracy. After finishing the description task, participants also completed a set of questionnaire items measuring demographic characteristics as well as degree of identification with their race and their gender. Ancillary measures like these identification questionnaires did not elicit clear and consistent effects across our studies and so are not reported here.

The eight figures used in the memory and description tasks were Black and white women and men in their twenties and thirties, wearing a variety of neutral clothing types. Each was assigned a monosyllabic heterogram name (Beth, Kate, Ruth, Jane, Fred, Dave, Mike and Luke) that was counterbalanced by race across lists. For the memory task, the figures were photographed separately standing still, and for the description task, the figures were photographed interacting directly with each other, illustrating the ten verbs/events: arguing, dancing, fighting, hugging, kissing, marrying, meeting, shouting, talking and touching.

To create the experimental stimuli for the description task, the photographed interactions were mirror-reversed and combined into animated GIFs using the Magick package in R[64], with the images reversing every 500 ms. Between experimental lists, we counterbalanced which figure began the animation on the left-hand side. The experimental protocol was created using the jsPsych library[65]. This mirror-reversing appeared to importantly reduce participants' bias to begin their description with the left-hand figure, which they did on only 54% (standard deviation (s.d.) 13%) of trials.

We pre-registered to exclude trials where response times were more than three s.d. from the mean, as well as trials on which participants used incorrect names or incorrect syntax, but to include trials on which participants made spelling mistakes. We pre-registered to exclude participants who took longer than 60 min to complete the study, who had more than 50% of their trials excluded or who did not provide the correct syntax on at least 50% of transitive and 50% of intransitive trials. We also pre-registered to exclude participants who showed a strong orientation bias, starting their description with either the initially left-hand-side figure on more than 90% or less than 10% of trials. Post hoc, we also excluded participants whose self-reported race did not match their Prolific records ($n = 5$) and who self-reported not speaking English as a first language despite this being part of their Prolific record ($n = 8$). The final sample comprised 196 participants (46 Black women, 45 Black men, 54 white women and 51 white men).

Our pre-registered analyses focused on analysing gender-bias trials and race-bias trials separately. In the main text, we typically report these trials together to provide a more holistic and powerful view of the data. The reported analyses thus depart from the pre-registration but follow the same basic logic and use the same technique: mixed-effects logistic regressions conducted using the package lme4[66]. All regressions had the structure (in lme4 syntax) LikeMe ~ 1 + syntax[transitive/intransitive] + (1 + syntax|subject) + (1 + syntax|event). Predictors were contrast coded.

### Experiment 2
The pre-registration for experiment 2 can be found at https://osf.io/t928m. This study used a new set of stimuli where the pictured figures were more standardized: matched for height, facial expression and wearing very similar clothing. To ensure standardization, we only used events in which the figures could be photographed separately and then the images combined (so that we could match for height). This meant that we had to replace verbs/events from experiment 1 in which the figures were physically touching, and so we substituted in the events 'playing cards' and 'flirting' for 'hugging' and 'meeting'.

A total of 240 new participants were recruited from Prolific.com, the same sample size as experiment 1 and using the same principles (60 white men, 60 Black men, 60 white women and 60 Black women, all aged over 18 and compensated £4). Participants from experiment 1 were excluded using Prolific's functionality. The procedure was matched except that the final questionnaire did not measure in-group identification but rather attitudes towards same-gender and interracial relationships. We used the same criteria for trial and participant exclusion as in experiment 1, resulting in a final sample of 195 participants (51 Black women, 42 Black men, 54 white women and 48 white men). We used the same regression models as experiment 1.

### Experiment 3
The pre-registration for experiment 3 can be found at https://osf.io/drb6f. This study was matched to experiment 2 but compared active transitive versus passive syntax. Since it was not possible to passivize all of the verbs used in experiment 2, we used a new set of ten verbs/events: hug, seduce, kiss, greet, look, touch, copy, challenge, oppose and shout.

We recruited 240 new participants from Prolific.com, using the same principles as experiments 1 and 2 (60 white men, 60 Black men, 60 white women and 60 Black women, all aged over 18 and compensated £4) except that all participants this time were required to be resident of the USA. We excluded participants and trials in the same fashion as before, but this time the pre-registration included exclusion of participants whose self-reported demographics differed from their Prolific-reported demographics. The final sample comprised 196 participants: 46 Black women, 49 Black men, 55 white women and 46 white men. We used the same regression models as experiment 1.

### Experiment 4
Experiment 4 was not formally pre-registered. Its design was based on similar follow-up studies conducted in our lab showing that, for

English speakers, a similarly sized like me effect can be found from stimuli like those in Fig. 3a, which allowed us to use a range of verbs beyond symmetric predicates, and minimized potential visual biases found in depictions of events. We used the same Black figures as experiments 2 and 3, and four new Asian figures (two male and two female) wearing similar clothing. For the Chinese condition, the figures were given homophonic translations of English names written using two characters (贝斯, 露丝, 凯特, 弗雷, 麦克, 卢克, 戴夫, 霍普 (the latter translates to the name Hope, which we used instead of Jane as the Chinese transliteration for Jane contains only one character)). Note that the names 霍普 (Hope) and 贝斯 (Beth) may be gender neutral in Chinese, which could cause observed like me gender effects to be smaller in that language. We used the verbs 'praising', 表扬; 'accompanying', 陪伴; 'walking towards', 走向; 'approaching', 接近; 'looking at', 看着; 'listening to', 聆听; 'contacting', 联系; 'introducing', 介绍; 'consulting', 咨询; and 'reminding', 提醒.

We aimed to recruit 60 female and 60 male native Chinese speakers (half assigned to the Chinese and half to the English condition). We recruited through a mixture of personal contacts and dark social media channels such as Whatsapp and WeChat groups, targeting students at UK universities. Our actual sample comprised 61 women and 61 men, all aged over 18 and compensated £5. Participants completed the study online.

Exclusion criteria were the same as experiment 3, except that there was no need to exclude participants on the basis of self- versus Prolific-reported demographic information. The final sample included 31 men and 27 women in the Chinese condition, and 28 men and 28 women in the English condition.

The regression models all had the structure LikeMe ~ 1 + language[Chinese/English] + (1|subject) + (1 + language|event) except for language-specific analyses that dropped the language comparison terms.

Note that, overall, Chinese- and English-language participants behaved similarly in this study, despite the latter using a second language. Accuracy was slightly worse for Chinese participants speaking English: 14% of English trials were excluded for making a mistake versus 13% for Chinese, and two Chinese-speaking-English participants were excluded for making too many mistakes versus one in Chinese. The median time taken to complete inputting the critical sentence was longer in the English condition (9.5 s) than the Chinese condition (8.3 s).

## Ratings studies

We ran two ratings studies, one measuring intuitions about agency and one measuring intuitions about speaker empathy and identification. For the agency studies, the transitive and conjoined subject versions of the rating study were separately pre-registered at https://osf.io/p9d5v/registrations and the passive version was pre-registered at https://aspredicted.org/7698v.pdf. Sixty-four participants (32 female, all aged over 18 and compensated £2.50) participated in each study using a between-subjects design. All reported that their first language was English, that their location was the UK and that they had no language-related disorders. All had a Prolific approval rating of between 95% and 100%, and none had taken part in our previous studies (excluded using Prolific's functionality).

On each test trial, participants read a sentence containing a symmetrical verb and two of the names used in experiments 1 through 4 (for example, 'Fred is playing with Ruth or Beth and Jane are playing with each other'). They then rated the relative agencies of the two characters in two ways: in terms of volition (is it more likely that Fred wanted to play with Ruth or Ruth wanted to play with Fred?) and in terms of causation (is it more likely that Fred asked to play with Ruth or Ruth asked to play with Fred?). These volition and causation dimensions of agency were chosen following previous linguistic work on what factors constitute agenthood[33,67]. Ratings were made on a Likert scale from 1 (Fred asked to play with Ruth) to 7 (Ruth asked to play with Fred). Across lists we

counterbalanced which names occurred in which positions in the critical sentence, and whether the subject of the critical sentence was anchored to 1 or 7 on the scale. Participants completed 80 test trials in total, 40 with different gender names and 40 with same gender names. They also completed 16 catch questions where the answers were not subjective (for example, critical sentence 'Ruth is chasing Fred' and answers 'Ruth is running from Fred' versus 'Fred is running from Ruth').

We pre-registered to exclude trials on which participants responded slower than 3 s.d. from the mean. We also pre-registered to exclude participants who answered more than two catch trials incorrectly, whose self-reported gender mismatched Prolific records and who reported not living in the UK for the last 5 years. Lastly, we pre-registered to exclude participants who used the same response on more than 80% of trials, but we removed that restriction for the final analysis as it had the consequence of excluding a large number of participants in the conjoined subject condition, who consistently reported that agenthood ratings were approximately the same between the two mentioned figures. The sample size after exclusions was 171 (87 female, 58 in the transitive 54 in the conjoined subject and 59 in the passive condition). Data were analysed using mixed-effects regressions of the form Agency Rating ~ 1 + Agency Property [volition/causation] + (1 + Agency Property|subject) + (1|verb) and Agency Rating ~ 1 + Grammatical Frame × Agency Property + (1 + Agency Property|subject) + (1|verb).

The empathy and identification rating study used a modified within-subjects design. Using the full set of verbs and frames used in experiments 1 through 3, 64 female and 64 male participants (all aged over 18 and compensated £2.50) each rated a mixture of active transitive (*n* = 19 trials), conjoined intransitive (*n* = 12 trials) and passives (*n* = 10 trials), along with the same fillers as the agency study. All reported that their first language was English and their location was the UK. None had taken part in our prior studies (implemented using Prolific's functionality).

On each trial, participants read a sentence of the form 'Somebody says "Luke and Beth are dancing with each other"', with the names determined by a random draw of those used in experiments 1 through 3. Participants then used a 1–7 scale to rate whether 'The speaker identifies more with Luke than Beth' or 'The speaker identifies more with Beth than Luke', with the left-to-right order of the response options randomized. We pre-registered to exclude trials on which participants respond slower than 3 s.d. from the mean, to exclude participants who answer more than two catch trials incorrectly, whose self-reported gender mismatched Prolific records or who reported not living in the UK for the last 5 years. Sample size after exclusions was 109 (58 female). Data were analysed using a pre-registered regression of the form Rating ~ Grammatical Frame + (1 + Grammatical Frame|Participant) + (1|Item).

## Multi-study analyses

We analysed the effect of syntax across the first three studies using a regression of the form LikeMe ~ 1 + Grammatical Frame + (1|subject) + (1|event).

We pre-registered our analysis of how demographic match affects naming speeds at https://aspredicted.org/jf9cn.pdf. This pre-registration was based on an initial exploratory analysis of the Chinese data in experiment 4, which showed qualitatively similar effects. Before analysis, we excluded all those participants who had been excluded for the analysis of sentence production. We also excluded all incorrect trials, all trials with response times longer than 10 s, and all follow-up trials (in which participants were offered a second shot at choosing the name for a figure following an initial incorrect response). The pre-registered analysis reported in the paper used a mixed-effects regression of the form log(Response Time) ~ Feature Match + (1 + Feature Match|id), where Feature Match is −1 if participant and figure match in both race and gender, 0 if they only match along one dimension and 1 if they match along neither dimension.

The main omnibus analyses used mixed-effects regressions of the form LikeMe ~ 1 + Participant Gender × Participant Race + (1|subject) + (1 + Participant Gender × Participant Race|event). Predictors were contrast coded, and separate models were fit to the race and gender trials. The analysis of trial type (using data from all four production studies) had the form LikeMe ~ 1 + Trial Type [Gender versus Race-focused] + (1 + Trial Type|subject) + (1 + Trial Type|event).

## Reporting summary

Further information on research design is available in the Nature Portfolio Reporting Summary linked to this article.

## Data availability

Stimuli and data for this paper can be found via Open Science Framework at https://osf.io/87vew/.

## Code availability

Code for this paper can be found via Open Science Framework at https://osf.io/87vew/.

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

## Acknowledgements

J.B. was funded by a PPLS PhD scholarship from the University of Edinburgh. H.R. was funded by grants ES/N005635/1 and ES/V012878/1 from the ESRC. The funders had no role in study design, data collection and analysis, decision to publish or preparation of the manuscript.

## Author contributions

J.B. developed the study concept with H.P.B. and H.R. J.B., L.T.H., H.P.B. and H.R. contributed to the study designs. J.B., S.H.W. and H.R. collected data. J.B. and H.R. conducted analyses. J.B., H.P.B. and H.R. contributed to the writing of the manuscript, and all authors edited and approved the final manuscript.

## Competing interests

The authors have no competing interests to declare.

## Additional information

**Correspondence and requests for materials** should be addressed to Jessica Brough.

# Reporting Summary

## Statistics

For all statistical analyses, confirm that the following items are present in the figure legend, table legend, main text, or Methods section.

| n/a | Confirmed | |
|---|---|---|
| ☐ | ☒ | The exact sample size (*n*) for each experimental group/condition, given as a discrete number and unit of measurement |
| ☐ | ☒ | A statement on whether measurements were taken from distinct samples or whether the same sample was measured repeatedly |
| ☐ | ☒ | The statistical test(s) used AND whether they are one- or two-sided *Only common tests should be described solely by name; describe more complex techniques in the Methods section.* |
| ☐ | ☒ | A description of all covariates tested |
| ☐ | ☒ | A description of any assumptions or corrections, such as tests of normality and adjustment for multiple comparisons |
| ☐ | ☒ | A full description of the statistical parameters including central tendency (e.g. means) or other basic estimates (e.g. regression coefficient) AND variation (e.g. standard deviation) or associated estimates of uncertainty (e.g. confidence intervals) |
| ☐ | ☒ | For null hypothesis testing, the test statistic (e.g. *F*, *t*, *r*) with confidence intervals, effect sizes, degrees of freedom and *P* value noted *Give P values as exact values whenever suitable.* |
| ☒ | ☐ | For Bayesian analysis, information on the choice of priors and Markov chain Monte Carlo settings |
| ☐ | ☒ | For hierarchical and complex designs, identification of the appropriate level for tests and full reporting of outcomes |
| ☐ | ☒ | Estimates of effect sizes (e.g. Cohen's *d*, Pearson's *r*), indicating how they were calculated |

*Our web collection on statistics for biologists contains articles on many of the points above.*

## Software and code

Policy information about availability of computer code

| Data collection | JSPsych library |
|---|---|
| Data analysis | R code using various libraries including dplyr and lme4 |

For manuscripts utilizing custom algorithms or software that are central to the research but not yet described in published literature, software must be made available to editors and reviewers. We strongly encourage code deposition in a community repository (e.g. GitHub). See the Nature Portfolio guidelines for submitting code & software for further information.

## Data

Policy information about availability of data

All manuscripts must include a data availability statement. This statement should provide the following information, where applicable:

- Accession codes, unique identifiers, or web links for publicly available datasets
- A description of any restrictions on data availability
- For clinical datasets or third party data, please ensure that the statement adheres to our policy

Stimuli, data, and code for this paper can be found at https://osf.io/87vew/.

# Research involving human participants, their data, or biological material

Policy information about studies with underline{human participants or human data}. See also policy information about underline{sex, gender (identity/presentation), and sexual orientation} and underline{race, ethnicity and racism}.

| | |
|---|---|
| Reporting on sex and gender | For our studies, we recruited participants who self-reported both their sex and their gender to be either male or female. We recruited equal numbers of each group. |
| Reporting on race, ethnicity, or other socially relevant groupings | For our studies, we recruited participants who self-reported as Black, white, or Asian. Note that participants reported race/ethnicity both directly to us and to the Prolific marketplace; if their reports different then these participants were excluded. |
| Population characteristics | Participants were recruited to match requirements for sex/gender characteristics, and race/ethnicity characteristics. We did not recruit based on age except that all participants were greater than 18 years of age. |
| Recruitment | Participants were recruited through the online marketplace Prolific, and via social media channels (particularly WeChat and Whatsapp) |
| Ethics oversight | University of Edinburgh |

Note that full information on the approval of the study protocol must also be provided in the manuscript.

# Field-specific reporting

Please select the one below that is the best fit for your research. If you are not sure, read the appropriate sections before making your selection.

☐ Life sciences     ☒ Behavioural & social sciences     ☐ Ecological, evolutionary & environmental sciences

For a reference copy of the document with all sections, see nature.com/documents/nr-reporting-summary-flat.pdf

# Behavioural & social sciences study design

All studies must disclose on these points even when the disclosure is negative.

| | |
|---|---|
| Study description | Cognitive psychological experiment generating behavioural data. |
| Research sample | Studies 1-3 use the Prolific participant pool, and recruited English speakers. Each of these studies recruited 60 Black men, 60 Black women, 60 white men and 60 white women. The Prolific participant pool is not demographically matched to the UK or US population, but is more representative than a standard University participant pool. Study 4 recruited Chinese students at UK Universities (60 male, 60 female) who spoke both Chinese and English and were tested in one of those languages. |
| Sampling strategy | Participants were sampled from either Prolific or communities as above. For Study 1, the sample size was set without a power analysis, based on the maximum number of participants we could reasonably recruit with the study budget. A subsequent power analysis using the procedures described in Westfall, Kenny & Judd, (2014) and the online application jakewestfall.shinyapps.io suggests that this sample size combined with our number of trials gives 80% power to detect an effect size of 0.23. Given the robust effects found in Study 1, we continued with that same sample size in subsequent studies. |
| Data collection | Participants took part in online psychological experiments in which they were shown words and images. |
| Timing | Data was collected between 2018 and 2023, in the order described in the paper. |
| Data exclusions | For Experiment 1, we pre-registered to exclude trials where response times were more than 3 standard deviations from the mean, as well as trials on which participants used incorrect names or incorrect syntax, but to include trials on which participants made spelling mistakes. We pre-registered to exclude participants who took longer than 60 minutes to complete the study, who had more than 50% of their trials excluded, or who did not provide the correct syntax on at least 50% of transitive and 50% of intransitive trials. We also pre-registered to exclude participants who showed a strong orientation bias, starting their description with either the initially-left-hand-side figure on greater than 90% or less than 10% of trials. Post-hoc, we also excluded participants whose self-reported race did not match their Prolific records (n=5) and who self-reported not speaking English as a first language despite this being part of their Prolific record (n=8). The final sample comprised 196 participants (46 Black women, 45 Black men, 54 white women, 51 white men). Subsequent Experiments used the same criteria, and detailed reports of that are given in the Methods section. |
| Non-participation | NA |
| Randomization | Participants were randomly allocated to different experimental lists. |

# Reporting for specific materials, systems and methods

We require information from authors about some types of materials, experimental systems and methods used in many studies. Here, indicate whether each material, system or method listed is relevant to your study. If you are not sure if a list item applies to your research, read the appropriate section before selecting a response.

## Materials & experimental systems

| n/a | Involved in the study |
|-----|----------------------|
| ☒ | Antibodies |
| ☒ | Eukaryotic cell lines |
| ☒ | Palaeontology and archaeology |
| ☒ | Animals and other organisms |
| ☒ | Clinical data |
| ☒ | Dual use research of concern |
| ☒ | Plants |

## Methods

| n/a | Involved in the study |
|-----|----------------------|
| ☒ | ChIP-seq |
| ☒ | Flow cytometry |
| ☒ | MRI-based neuroimaging |

