## [Peer Review File · Nature Human Behaviour]

Peer Review Information

Journal: Nature Human Behaviour

Manuscript Title: Cognitive causes of Like Me race and gender biases in human language production

Corresponding author name(s): Jessica Brough

Editorial Notes:

Reviewer Comments & Decisions:

Decision Letter, initial version:

31st August 2023

Dear Dr Brough,

Thank you once again for your manuscript, entitled "Accessibility and incrementality cause "Like Me" race and gender biases in human language production", and for your patience during the peer review process.

Your Article has now been evaluated by 4 referees. You will see from their comments copied below that, although they find your work of considerable potential interest, they have raised some substantial concerns. In light of these comments, we cannot accept the manuscript for publication, but would be interested in considering a revised version if you are willing and able to fully address reviewer and editorial concerns.

We hope you will find the referees' comments useful as you decide how to proceed. If you wish to submit a substantially revised manuscript, please bear in mind that we will be reluctant to approach the referees again in the absence of major revisions. We are committed to providing a fair and constructive peer-review process. Do not hesitate to contact us if there are specific requests from the reviewers that you believe are technically impossible or unlikely to yield a meaningful outcome.

In particular, we note that Reviewer #1 and Reviewer #4 both feel that the current experiments do not definitively establish an accessibility mechanism for the current results. Therefore, we ask that you carry out additional - preregistered - experiment(s) to provide compelling evidence that the mechanism of the Like Me effect is due to accessibility. If you would like to discuss this request further, please email me at **[REDACTED]**

If you wish to submit a suitably revised manuscript, we would hope to receive it within 4 months. I would be grateful if you could contact us as soon as possible if you foresee difficulties with meeting

this target resubmission date.

- Include a "Response to the editors and reviewers" document detailing, point-by-point, how you addressed each editor and referee comment. If no action was taken to address a point, you must provide a compelling argument. When formatting this document, please respond to each reviewer comment individually, including the full text of the reviewer comment verbatim followed by your response to the individual point. This response will be used by the editors to evaluate your revision and sent back to the reviewers along with the revised manuscript.
- Highlight all changes made to your manuscript or provide us with a version that tracks changes.

[REDACTED]

Thank you for the opportunity to review your work. Please do not hesitate to contact me if you have any questions or would like to discuss the required revisions further.

Sincerely,
[REDACTED]

Reviewer expertise:

Reviewer #1: English linguistics

Reviewer #2: Online recruitment of different demographic groups

Reviewer #3: Chinese linguistics

Reviewer #4: Sociolinguistics

REVIEWER COMMENTS:

Reviewer #1:
Remarks to the Author:

The authors find that the order in which people describe two participants engaging in symmetric joint actions (hugging, touching) is influenced by who the participants are: female participants are more likely to produce the names of females first, black speakers are more likely to name the black person first, white speakers are more likely to name the white person first—even though, in the race manipulations, the same names were used, ensuring that the effect is not due to the frequencies of the names. The Results of 5 preregistered studies are appropriately analyzed and are clearly explained. The paper is likely to be of very broad interest, and I strongly recommend it be published with minor revisions.

I believe the paper can make a deeper point: The authors discuss the possibility that the Like-Me effect is due to the construal of in-group members as being more agentive (pg. 5-7) but argue instead that it is due to accessibility. The argument rests on the production of passive sentences (Experiment 3), in which the Like-Me effect is evident even though the first-named-entity is the undergoer. But Exp 3 does not address the possibility that Like-Me effect is due to empathy or perspective rather than agentivity, as argued explicitly by Kuno & Kaburaki (1977).

A stronger argument in favor of accessibility (perhaps mediated by empathy in the current experiments) might use arguments in Goldberg & Lee (2021), who proposed that accessibility explains the word order of all conjoined phrases, noting that more accessible word order choices become conventionalized if recurrent and not-meaning changing and then can attract similar cases. Mention of conventionalization would serve to motivate the authors' initial discussion of the fact that conventional language comes to reflect general cultural biases.

This would then relate the current work to the wealth of evidence of biases in language, from "traditional social psychological and sociolinguistic accounts" instead of claiming that the current work is distinct from such findings (middle PP page 3). That is, accessibility is not at odds with the emergence of biases in conventional language; it provides the psychological mechanism that gives rise to such biases.

Cooper & Ross (1975) had actually proposed an insightful if somewhat tongue-in-cheek "Me First principle," observing that in conjunctions, the male entity is conventionally produced first (king and queen); alcoholic comes first (gin and soda), large first (large and small), etc. I don't have strong feelings about whether Me First or Like Me is a better term, but a more prominent mention of this as a precedent for the current findings seems warranted.

To be clear, the authors' work goes well beyond any prior work I know of in offering empirical data (cf. Kuno & Kaburaki 1977; Cooper & Ross 1975) and by considering conjoined subjects, transitive subjects, and passive subjects (cf. rather than only conjunctions or only transitives/passives), and by finding the effect within men, women, black, white, and Asian groups as well as regardless of whether Chinese or English is used.

The work makes an important contribution to the literature and is likely to be widely read and cited.

(Currently uncited)

Goldberg, A. E., & Lee, C. (2021). Accessibility and historical change: An emergent cluster led uncles and aunts to become aunts and uncles. *Frontiers in psychology*, 12, 662884.

Kuno, S., & Kaburaki, E. (1977). Empathy and syntax. *Linguistic inquiry*, 627-672.

Signed
Adele Goldberg

Reviewer #2:

Remarks to the Author:

I have been asked to provide a review only on the use of an online platform (Prolific Academic) to recruit most of the participants across different demographic group. Based on the information in the Methods section, Prolific was used in Experiments 1-3. It was also used in the "Ratings Study" but not to sample across demographic groups so I will not relate to it. In experiments 1-3, the authors aimed to recruit 60 White/Black Men/Women. They used the prescreeners available on Prolific to sample from these populations for each experiment. I think that Prolific is capable of providing these sample sizes and probably more than that. As of today, Prolific's interface states there are about 12,000

eligible participants on Prolific that are said to be active in the last 90 days. Among those, about 8,000 are Female. Applying the English as First Language additional screener still shows more than 3,000 eligible participants. So, I would say that it is reasonable to expect recruiting about 60 of each of the above groups for each experiments using Prolific. In my experience, the prescreening data is reliable as any other platform or panel that I have used or studied.

I'd like to add two additional comments related to that, however, concerning repeated participation and power. The authors say they sample "new" participants in each study, but I think they should be more clear to state how they actually blocked participants from previous studies: did they use that option on Prolific, another feature on their survey software, checked for repeated IPs? Secondly, the authors state that their sample size provides 80% power to detect $d > 0.23$. This may be true for a sample of 240 in total, but could not be possible for sub-samples of 60 each. I'd suggest the authors conduct a power sensitivity analysis using G*Power or a similar software to double-check their calculations.

Other than that, I have not reviewed, as I was requested, the other parts of the paper and I wish the authors best of luck in their research.

Reviewer #3:

Remarks to the Author:

This is a review for the manuscript "Accessibility and incrementality cause "Like Me" race and gender biases in human language production". This paper reports a series of four experiments, which were designed to investigate whether, other things being equal, speakers would tend to name an individual who is more similar to the speaker (in gender or ethnicity) than one that is less similar when describing an event that involve both individuals.

My review will focus on Experiment 4, the one with Chinese stimuli and Chinese-speaking participants, which is also the only one experiment reported in this paper that was not pre-registered. Overall, I think the design, implementation and data analysis of Experiment 4 is solid, but I do have some questions and comments, as listed below:

- First, how was the memory task conducted? Were the East Asian characters introduced as "Asians" (or "亚洲人" in Chinese) or "Chinese" ("中国人" in Chinese)? Labeling the Asian characters as "Chinese" should safely give these characters the "in-group" status with Chinese participants, but I'm not sure if labeling them as "Asian" would achieve the same "in-group" status, because the "Asian" label could imply non-Chinese Asians to Chinese participants under the maxim of quantity (e.g., if they are Chinese, they would have been referred to as "Chinese" instead of "Asian"). Also, in the Chinese condition, depending on which label is used for "Asians" (e.g., "亚裔" "Asian descent", or "亚洲人" "Asian") or "Chinese" (e.g. "华裔" "Chinese descent", or "中国人" "Chinese nationals"), there could be subtle differences as to whether the label refers to overseas Asians/Chinese or homeland Asians/Chinese, which could change the membership status of the characters depending on the Chinese participants' own Asian identity. Of course, one can still make the argument that the Chinese participants are ethnically closer to the East Asian characters than to the Black characters, so this wouldn't change the conclusion of Experiment 4. But more clarity about the methods would be good.

- Second, among the character names used in the Chinese condition, there are at least two, 贝斯 and 霍普, that may not be highly gendered marked for Chinese speakers who use English as an additional language. The name 霍普 (Hope) may be gender-neutral in both the Chinese condition and the English condition for Chinese speakers, simply because it's probably not a name that Chinese speakers are familiar with. The name 贝斯 (Beth) may be gender-neutral to Chinese speakers in the Chinese condition (but probably not in the English condition), mainly due to the use of 斯 in the Chinese-translated name. If some of the names are gender-neutral to the Chinese participants, the observed

bias may be an underestimation of the actual bias in the gender-related trials.

- Third, I'm a bit surprised by the verb frames used in Experiment 4, since they are quite different from those used in the other experiments. The verbs in Experiment 4 are all asymmetrical. From the illustration in Fig 2 of the sentence "霍普正在接近凯特" (Hope is approaching Kate), I don't know how participants could tell who is approaching whom. In fact, on a side note, is the illustration in Fig 2 supposed to show a gender trial (i.e., Kate being a girl and Hope being a guy, and both are Black?)? I'm confused. I would like to see more examples of what the visual illustrations look like for different verbs in Experiment 4. Furthermore, I'd also like to see a more detailed description of the instruction given to the participants in Experiment 4. Relatedly, since the verbs are all asymmetrical, potentially participants could make mistakes if they mixed up the two characters in the response sentence. Does this happen? How often?

- The authors report that the magnitude of the Like Me effect doesn't vary between the Chinese condition and the English condition in Experiment 4. Apart from effect size, do the participants show any other differences between the native language and the nonnative language? For example, is there an accuracy difference between the two conditions?

In addition, I have a few more questions and comments about the paper in general:

- According to Figure 3, there seems to be a general trend for female participants to show less "Like Me" effect than male participants in the gender trials. Is this pattern significant? The authors mention that Black Women are the only group with no significant Like Me effect in the gender trials, but maybe a more general pattern is that men show a greater Like Me effect in gender trials than women, because of the addition of the Like Me effect and the androcentric effect?

- p9, second paragraph. Following from the preceding paragraph, I thought the analysis described in this paragraph are comparisons across Experiment 1 through 3. But the discussion in the second paragraph also mentions Asian participants, who only participated in Experiment 4. Please clarify.

Reviewer #4:

Remarks to the Author:

Thank you for letting me review the manuscript "Accessibility and incrementality cause "like me" race and gender biases in human language production".

In 4 preregistered experiments, the authors test and find support for a competing theory about which representative of a social group that is presented first in sentences construction. They call this effect the "like-me"-effect indicating that people present a person that are more similar to them than the other person. For example, a white women mentions a white person rather than a black person first, and a woman rather than a man first. The authors claim that the effect origins from a cognitive, automatic retrieval mechanism rather than social aspects related with agency and status. I agree that researchers should always try to find alternative reasons for how and why people present different social categories and what effects presentation order might have since it can affect status and norms perceptions in societies. However, I am not fully convinced how alternative the findings of these four studies are.

Past research has focused on presentations of groups (e.g., men vs women, black vs white). In these studies, people learn to know about individuals and their category belongings. The authors argue that people don't have other groups as accessible in memory and that they are less accurate in recognition of outgroup members which would slow down how quick these other groups come to mind. In the completed studies, the participants learn about the category that each individual represent to ceiling effects. In this way, I did not follow how the effects could be attributed to automatized mechanisms in which outgroups are harder to represent accurately as participants have shown that they can do that.

The authors focus on social biases related with agency or with status in societies and claim that for example androcentrism should not account for the results, or that the agentive object of a sentence would be a way to disentangle whether this affect occurs or not. However, I do understand why fundamental biases like self-serving perspectives could not account for the results that the authors find. What makes their results purely about automacy and retrieval ease? As the authors make a strong case about cognitive rather than social factors that accounts for the results, the tests and the arguments need to be clearer.

In relation to Nature Human behaviors guidelines on significant advance I cannot see that this paper presents such novelty to be published in this journal. I really do appreciate the paper and find that it provides some conceptual novelty in how individuals from different groups are presented, but more research is needed that provides stronger evidence for the "like-me" hypothesis. Thus, it is the novelty in relation to past literature on self-serving biases in combination how well the different studies can disentangle social effects from cognitive effects that brings me to a conclusion to not endorse publication of the present paper.

Author Rebuttal to Initial comments

Response to Reviews Reviewer

1

The authors find that the order in which people describe two participants engaging in symmetric joint actions (hugging, touching) is influenced by who the participants are: female participants are more likely to produce the names of females first, black speakers are more likely to name the black person first, white speakers are more likely to name the white person first—even though, in the race manipulations, the same names were used, ensuring that the effect is not due to the frequencies of the names. The Results of 5 preregistered studies are appropriately analyzed and are clearly explained. The paper is likely to be of very broad interest, and I strongly recommend it be published with minor revisions.

I believe the paper can make a deeper point: The authors discuss the possibility that the Like-Me effect is due to the construal of in-group members as being more agentive (pg. 5-7) but argue instead that it is due to accessibility. The argument rests on the production of passive sentences (Experiment 3), in which the Like-Me effect is evident even though the first-named-entity is the undergoer. But Exp 3 does not address the possibility that Like-Me effect is due to empathy or perspective rather than agentivity, as argued explicitly by Kuno & Kaburaki (1977).

Many thanks for this point – we now directly address it throughout the manuscript, including with new data. The Reviewer’s argument is based on an important analysis from theoretical syntax, which claims that grammars are biased to have constructions in which the person with whom we most identify is placed first in

the sentence, typically as the grammatical subject. As the Reviewer notes later, there is actually a chicken-and-egg problem here. Kuno & Kaburaki's proposal does not explain *why* the empathised person ought to go first, but rather notes it as a generalisation. Our proposal can actually explain why languages have evolved to show this generalisation (and we now highlight this in the Discussion).

Still, we also think that the two phenomena can be disentangled, even though we agree with the Reviewer that they are likely interrelated. Specifically, we can evaluate whether empathy specifically causes the Like Me bias by characterising intuitions about empathy across our stimuli, and evaluating whether they track the size of the Like Me bias.

The logic here is similar to the evaluations of agency that we described in our original manuscript. In that study, we measured participants' intuitions as to whether the *agency* of the first mentioned person in a sentence was greater than the second mentioned person. The difference was particularly marked for the transitive sentences (*Kate is kissing Dave*) but also held for the conjoined subject sentences (*Kate and Dave are kissing each other*), although to a much smaller degree. This small difference motivated us to run the passive experiment, in which the first mentioned person is no longer the agent – although they may receive more empathy, as the Reviewer points out.

We therefore collected a new dataset in which participants judged the communicated empathy of the stimuli used in our study, and evaluated if this predicted the size of the Like Me bias. Participants were shown sentences in which an unknown speaker describes an interaction between two people. They then had to rate whether this unknown speaker identifies more with the first person mentioned or the second person mentioned (a reasonable operationalisation of Kuno & Kaburaki's empathy construct). For example, they might read

Somebody says

Luke is being talked to by Beth

And then rate whether the speaker identifies more with Beth or more with Luke, using a 1-7 scale.

As one might expect, we found that empathy intuitions strongly varied across the constructions used in our stimuli. Empathy for the first mentioned person was highest for passive sentences, lowest for conjoined subject intransitives (*Luke and Beth are talking to each other*) and in the middle for active transitives (*Luke is*

talking to Beth).

Importantly, we also found that this factor does not predict the size of the Like Me effect. Instead, the Like Me bias is invariant across constructions. We therefore suggest that the act of communicating empathy and social identity does not cause the Like Me effect.

A stronger argument in favour of accessibility (perhaps mediated by empathy in the current experiments) might use arguments in Goldberg & Lee (2021), who proposed that accessibility explains the word order of all conjoined phrases, noting that more accessible word order choices become conventionalized if recurrent and not-meaning changing and then can attract similar cases. Mention of conventionalization would serve to motivate the authors' initial discussion of the fact that conventional language comes to reflect general cultural biases.

Thank you for this! Note that our prior evidence for accessibility also came from the null comparison between active transitive (*Dave is dancing with Kate*) and conjoined subject conditions (*Dave and Kate are dancing*) in Experiments 1 and 2. This accords well with the Reviewer's suggestion that accessibility may be a key predictor of which conjoined phrases are likely to conventionalise. Unfortunately we struggled to integrate this into the Introduction without significantly reframing the paper, but we have added the point in the Discussion, as a way to motivate the broader linguistic implications of our findings.

Moreover, we now include new analyses that provide an even stronger empirical case for an accessibility account of the Like Me effect. Specifically we show that participants in our task were also faster to access (i.e., retrieve from memory) the names of people more like them. In our original experiments, participants had to learn the names of eight characters and then were repeatedly tested on these throughout the study. We pre-registered a secondary analysis of these data and show that participants overall were fastest to report the names of people who matched them in both gender and race, were about 75ms slower to report the names of people who matched them on only one dimension, and were a further 50ms slower to report the names of people who did not match them on either gender or race. Thus, given that retrieval speed is slower for these names, we would naturally expect accessibility-driven ordering effects in sentence production.

This would then relate the current work to the wealth of evidence of biases in language, from "traditional social psychological and sociolinguistic accounts" instead of claiming that the current work is distinct from such findings (middle PP page 3).

That is, accessibility is not at odds with the emergence of biases in conventional language; it provides the psychological mechanism that gives rise to such biases.

Cooper & Ross (1975) had actually proposed an insightful if somewhat tongue-in-cheek “Me First principle,” observing that in conjunctions, the male entity is conventionally produced first (king and queen); alcoholic comes first (gin and soda), large first (large and small), etc. I don’t have strong feelings about whether Me First or Like Me is a better term, but a more prominent mention of this as a precedent for the current findings seems warranted.

Thank you for this. We had cited this paper in the first paragraph, but agree that its influence deserves heavier weight. To be clear to the Editor, Cooper & Ross’ important work argued that conventionalised phrases (like *king and queen*) were the product of a slightly different bias, a *metaphorical* Me First bias in which the Me in question is not the individual speaker, but rather an imagined prototypical speaker of the language. Cooper and Ross construed this speaker as typically male, powerful, large, etc, and this is why we say *king and queen* and *brother and sister* rather than the reverse.

Our work goes importantly beyond that in suggesting that the bias is not metaphorical but a real aspect of language in the mind, showing that the bias does not refer to a prototypical speaker but rather to the actual speaker themselves, and showing that the bias affects even sentences that are completely matched (e.g., as when we counterbalanced names for figures of different races). From there, the Reviewer’s point is well taken that this individual-level bias could scale up over cultural evolution to generate the data that Cooper and Ross analysed. We now note our debt to that paper more explicitly in the manuscript (e.g., *We term this potential bias the Like Me effect, building on prior claims about historical and conventionalised social biases in word order*²⁰) **and highlight how it links to our results in the Discussion.**

To be clear, the authors’ work goes well beyond any prior work I know of in offering empirical data (cf. Kuno & Kaburaki 1977; Cooper & Ross 1975) and by considering conjoined subjects, transitive subjects, and passive subjects (cf. rather than only conjunctions or only transitives/passives), and by finding the effect within men, women, black, white, and Asian groups as well as regardless of whether Chinese or English is used.

The work makes an important contribution to the literature and is likely to be widely read and cited.

Thank you for this!

(Currently uncited)

Goldberg, A. E., & Lee, C. (2021). Accessibility and historical change: An emergent cluster led uncles and aunts to become aunts and uncles. *Frontiers in psychology*, 12, 662884.

Kuno, S., & Kaburaki, E. (1977). Empathy and syntax. *Linguistic inquiry*, 627-672.

Signed

Adele Goldberg

Reviewer #2:

Remarks to the Author:

I have been asked to provide a review only on the use of an online platform (Prolific Academic) to recruit most of the participants across different demographic group. Based on the information in the Methods section, Prolific was used in Experiments 1-3. It was also used in the "Ratings Study" but not to sample across demographic groups so I will not relate to it. In experiments 1-3, the authors aimed to recruit 60 White/Black Men/Women. They used the prescreeners available on Prolific to sample from these populations for each experiment. I think that Prolific is capable of providing these sample sizes and probably more than that. As of today, Prolific's interface states there are about 12,000 eligible participants on Prolific that are said to be active in the last 90 days. Among those, about 8,000 are Female. Applying the English as First Language additional screener still shows more than 3,000 eligible participants. So, I would say that it is reasonable to expect recruiting about 60 of each of the above groups for each experiments using Prolific. In my experience, the prescreening data is reliable as any other platform or panel that I have used or studied.

Thank you for this. We found that Prolific was very suitable for our methods but, as the Reviewer suggests, the Prescreening was not 100% reliable and so, as noted in our manuscript, we did exclude participants who reported different characteristics in our survey versus their Prolific pre-screening.

I'd like to add two additional comments related to that, however, concerning repeated participation and power. The authors say they sample "new" participants in each study, but I think they should be more clear to state how they actually blocked participants from previous studies: did they use that option on Prolific, another feature on their

survey software, checked for repeated IPs?

We used the functionality provided by Prolific to hide each new study from account holders who had taken part in the prior studies, thereby automatically avoiding any repeated participation across both the language production and the rating studies. We have considered that users of Prolific might make multiple accounts under different identifiers in order to take part in the same study more than once, however we feel reassured by the measures Prolific takes (which you can read more about here) to avoid this type of behaviour, and do not consider this to be a concern. This includes IP address detection. Consequently, we are confident that those who participated in Experiment 1 did not also participate in the latter experiments in this paper, and so on.

In the Methods section we now note that participants were excluded using Prolific's functionality.

Secondly, the authors state that their sample size provides 80% power to detect $d > 0.23$. This may be true for a sample of 240 in total, but could not be possible for sub-samples of 60 each. I'd suggest the authors conduct a power sensitivity analysis using G*Power or a similar software to double-check their calculations.

Thank you for this. For the sub-samples of 60 each, our power is still quite high because each participant completed 80 items. We implemented this using Westfall et al's (2014) method [see https://jakewestfall.shinyapps.io/two_factor_power/] and found that for 60 participants we had 80% power to detect an effect of size 0.31. We now state this clearly in the Methods section.

Other than that, I have not reviewed, as I was requested, the other parts of the paper and I wish the authors best of luck in their research.

Reviewer #3:

Remarks to the Author:

This is a review for the manuscript "Accessibility and incrementality cause "Like Me" race and gender biases in human language production". This paper reports a series of four experiments, which were designed to investigate whether, other things being equal, speakers would tend to name an individual who is more similar to the speaker (in gender or ethnicity) than one that is less similar when describing an event that

involve both individuals.

My review will focus on Experiment 4, the one with Chinese stimuli and Chinese-speaking participants, which is also the only one experiment reported in this paper that was not pre-registered. Overall, I think the design, implementation and data analysis of Experiment 4 is solid, but I do have some questions and comments, as listed below:

Thank you for this!

- First, how was the memory task conducted? Were the East Asian characters introduced as "Asians" (or "亚洲人" in Chinese) or "Chinese" ("中国人" in Chinese)? Labeling the Asian characters as "Chinese" should safely give these characters the "in-group" status with Chinese participants, but I'm not sure if labeling them as "Asian" would achieve the same "in-group" status, because the "Asian" label could imply non-Chinese Asians to Chinese participants under the maxim of quantity (e.g., if they are Chinese, they would have been referred to as "Chinese" instead of "Asian").

Apologies, our writing was unclear. We never introduced the characters in terms of their ethnicities. We simply told participants that they would be learning the names for, and talking about, 8 figures, and then showed them their pictures and names. Half of the figures were Asian and half were Black but their ethnicity was never explicitly stated. Thus, we relied on the visual stimulus to give the characters in-group status. We now aim to make this clear earlier in the paper, in Experiment 1, where we write *No further social information about the figures was provided beyond their photograph and name. In the description of Experiment 4 we write *Participants learned the names for the figures in the same way as before, and again no background information was given about the figures beyond their names and photographs.**

Also, in the Chinese condition, depending on which label is used for "Asians" (e.g., "亚裔" "Asian descent", or "亚洲人" "Asian") or "Chinese" (e.g. "华裔" "Chinese descent", or "中国人" "Chinese nationals"), there could be subtle differences as to whether the label refers to overseas Asians/Chinese or homeland Asians/Chinese, which could change the membership status of the characters depending on the Chinese participants' own Asian identity. Of course, one can still make the argument that the Chinese participants are ethnically closer to the East Asian characters than to the Black characters, so this wouldn't change the conclusion of Experiment 4. But more clarity about the methods would be good.

As stated above, we have edited the manuscript to make clear that the groups

were only minimally presented and thus ensure that other readers do not get confused in the same way as we confused the reviewer.

- Second, among the character names used in the Chinese condition, there are at least two, 贝斯 and 霍普, that may not be highly gendered marked for Chinese speakers who use English as an additional language. The name 霍普 (Hope) may be gender-neutral in both the Chinese condition and the English condition for Chinese speakers, simply because it's probably not a name that Chinese speakers are familiar with. The name 贝斯 (Beth) may be gender-neutral to Chinese speakers in the Chinese condition (but probably not in the English condition), mainly due to the use of 斯 in the Chinese-translated name. If some of the names are gender-neutral to the Chinese participants, the observed bias may be an underestimation of the actual bias in the gender-related trials.

Thank you for this point, which we have incorporated into the manuscript. As the Reviewer notes, this is not a confound because, if anything, it ought to reduce the size of the Like Me bias.

We now note this in the main manuscript and in the Methods section we write: *Note that the names 贝斯霍普 (Hope) and 贝斯 (Beth) may be gender-neutral in Chinese, which could cause observed Like Me gender effects to be smaller in that language.*

- Third, I'm a bit surprised by the verb frames used in Experiment 4, since they are quite different from those used in the other experiments. The verbs in Experiment 4 are all asymmetrical. From the illustration in Fig 2 of the sentence "霍普正在接近凯特" (Hope is approaching Kate), I don't know how participants could tell who is approaching whom.

First, an apology – the example sentence response in Figure 2 is wrong and is corrected in the Revision. It should have read 露丝正在接近卢克. In a previous draft of the manuscript, the figure showed a Black female (i.e., Kate) and Asian female character (i.e., Hope). However, that Asian individual did not give permission for their likeness to be reproduced in the pages of the journal, and so we cannot use their image. We therefore swapped the figure to a gender trial where the two

depicted individuals had given their permission for their photos to be used, but we neglected to update the example response. That has now been changed.

Second, some clarification because our writing was again unclear: This experiment used a different method from the three previous ones, a method which we had used in other follow up studies in our group and so were confident would generate a Like Me effect. Unlike in Experiments 1 through 3, participants here no longer saw a depicted event involving two characters. Instead, they simply saw the two individuals side-by-side, and then saw a sentence fragment that they had to complete. At that point, therefore, they had to implicitly decide which of the two individuals was going to play which role in the sentence. The strong advantage of this method is that it allows us to use a wider range of verbs, including the asymmetric ones discussed here, and verbs with different semantic and emotional valences that we have explored in other follow-up projects. Thus, participants did not have to perceive who was approaching whom, rather they had to imagine who was approaching whom in generating their description.

We now write: We used a slightly modified experimental design in which participants no longer described two figures interacting in an event, but instead were shown a mirror-reversing pair of figures standing side-by-side in a non-interactive fashion. Then, participants were given a sentence frame, and finally they produced a sentence using the figures' names and the frame (Figure 2A). This design (which pilot work showed elicited a Like Me effect in English) therefore differed from Experiments 1 through 3 in that we did not depict events directly, but rather allowed participants to imagine them. This meant that we could now use a distinct set of non-symmetric verbs as stimuli, and thus further assess generalisation across linguistic materials.

In fact, on a side note, is the illustration in Fig 2 supposed to show a gender trial (i.e., Kate being a girl and Hope being a guy, and both are Black?)? I'm confused. I would like to see more examples of what the visual illustrations look like for different verbs in Experiment 4. Furthermore, I'd also like to see a more detailed description of the instruction given to the participants in Experiment 4. Relatedly, since the verbs are all asymmetrical, potentially participants could make mistakes if they mixed up the two characters in the response sentence. Does this happen? How often?

We hope that our response above has clarified these points, but for completeness:

- 1. One of the names here was not correct in the original ms (but has now been corrected). The visual illustrations all looked like those in the Figure: two people standing side-by-side.**

2. **Our OSF repository, linked in the main paper, contains all of our stimuli for anybody to view.**
3. **Below, we show the infographics that participants were shown as instructions to explain the study methods (Examples for Study 1 and the Chinese condition of Study 4). We have not included these in the final paper because of concerns about space, but they are in the OSF repository.**
4. **Participants could not mix up the two characters because the participants themselves had to choose their roles.**

[FIGURE REDACTED]

- The authors report that the magnitude of the Like Me effect doesn't vary between the Chinese condition and the English condition in Experiment 4. Apart from effect size, do the participants show any other differences between the native language and the nonnative language? For example, is there an accuracy difference between the two conditions?

Thank you for this suggestion. In fact, our participants did not behave that differently across the languages. Accuracy in Experiment 4 was slightly worse for Chinese participants producing their non-native language (i.e., English): 14% of English trials were excluded for making a mistake versus 13% for Chinese trials, and two participants producing English were excluded for making too many mistakes versus one producing Chinese. But generally we had to exclude very few of these participants, perhaps because they were recruited by word of mouth rather than a system like Prolific, and so were more engaged in the task. That being said, the median time taken to complete inputting the critical sentence was greater in the English condition (9.5s) as compared to the Chinese condition (8.3s), indicating that the task was harder. We now describe this in the methods section.

In addition, I have a few more questions and comments about the paper in general:

- According to Figure 3, there seems to be a general trend for female participants to show less "Like Me" effect than male participants in the gender trials. Is this pattern significant? The authors mention that Black Women are the only group with no significant Like Me effect in the gender trials, but maybe a more general pattern is that men show a greater Like Me effect in gender trials than women, because of the addition of the Like Me effect and the androcentric effect?

Thank you for this suggestion. We were also interested in whether there might be a general androcentric effect, and so we reanalysed our data using the same

regression structure but a different regression contrast coding scheme that set the reference level as average performance across groups (rather than having one demographic group as the reference level). Under this analysis, we now see an overall gender effect, with women showing a smaller Like Me effect than men. There is still an interaction with race, however, with Black women showing much the smallest Like Me gender effect. We now write *Again, we found that a Like Me effect held overall (Beta=0.21(0.02), z=8.5, p<.001), but it was significantly smaller in women than in men (Beta=-0.2(0.05), z=4.2, p<.001). Moreover, and importantly, an interaction between gender and race indicated that the size of the Like Me gender effect was even smaller in Black women (Beta=-0.28(0.09), z=3.0, p=.003).*

- p9, second paragraph. Following from the preceding paragraph, I thought the analysis described in this paragraph are comparisons across Experiment 1 through 3. But the discussion in the second paragraph also mentions Asian participants, who only participated in Experiment 4. Please clarify.

Apologies for the poor writing here. We have made clear that all participants were included.

Reviewer #4:

Remarks to the Author:

Thank you for letting me review the manuscript “Accessibility and incrementality cause “like me” race and gender biases in human language production”.

In 4 preregistered experiments, the authors test and find support for a competing theory about which representative of a social group that is presented first in sentences construction. They call this effect the “like-me”-effect indicating that people present a person that are more similar to them than the other person. For example, a white women mentions a white person rather than a black person first, and a woman rather than a man first. The authors claim that the effect origins from a cognitive, automatic retrieval mechanism rather than social aspects related with agency and status. I agree that researchers should always try to find alternative reasons for how and why people present different social categories and what effects presentation order might have since it can affect status and norms perceptions in societies. However, I am not fully convinced how alternative the findings of these four studies are.

Past research has focused on presentations of groups (e.g., men vs women, black vs white). In these studies, people learn to know about individuals and their category

belongings. The authors argue that people don't have other groups as accessible in memory and that they are less accurate in recognition of outgroup members which would slow down how quick these other groups come to mind. In the completed studies, the participants learn about the category that each individual represent to ceiling effects. In this way, I did not follow how the effects could be attributed to automatized mechanisms in which outgroups are harder to represent accurately as participants have shown that they can do that.

Thank you for this comment, and we are happy to expand on our logic. In our task, we did not ask participants to learn about the categories that each character represents. Instead, we just asked our participants to learn names for each character: a Black man called *Mike*, a white woman called *Beth*, another Black man called *Fred*, a Black woman called *Ruth*, etc. Categories were always left implicit, defined only by the obvious race and gender of the character. Our hypothesis is that the accessibility of each character's names in memory will then be influenced by the degree of match between the social categories of the character and those of the participant.

There are a number of factors that could cause social category match to influence accessibility, including perceptual factors such as greater expertise at recognising the faces of members of your own social group, and more cognitive/conceptual factors such as predicability. Moreover, and directly relevant to the reviewer's final point, differences in accessibility still exist for automatised responses. For example, the accessibility of a word depends on its frequency, such that a more frequent word like *dog* is more accessible than a matched but less-frequent word like *owl*. This means that even if participants had learned all the names to ceiling, it can still be easier [i.e., faster] for them to bring to mind the names of in-group members as compared to out-group members. We return to this point momentarily, because we now have direct evidence for it from within our own paradigm.

The authors focus on social biases related with agency or with status in societies and claim that for example androcentrism should not account for the results, or that the agentic object of a sentence would be a way to disentangle whether this affect occurs or not. However, I do understand why fundamental biases like self-serving perspectives could not account for the results that the authors find. What makes their results purely about automacy and retrieval ease? As the authors make a strong case about cognitive rather than social factors that accounts for the results, the tests and the arguments need to be clearer.

Thank you for this point. Our original argument for accessibility was based upon the fact that the bias was invariant across grammatical constructions. By contrast, an explanation based on self-serving perspectives would predict that the size of the Like Me bias should differ when the grammatical frame of a sentence communicates different facets and perspectives. For instance, the subject of active transitive sentences (*Luke is dancing with Ruth*) would presumably be a highly self-serving position, since the sentence is about the subject, and the subject is highlighted as the agent of the event. By contrast, the first named person in a conjoined subject (*Luke and Ruth are dancing with each other*) would not be particularly self-serving, since the sentence is not simply about the first named person, and that first named person is only weakly highlighted as the more agentive individual. Instead, we find that the Like Me effect is the same across structures. This is consistent with an explanation based on accessibility.

However, we recognise that our original paper did not include strong positive evidence for the accessibility account. Our revision now provides that. In particular, we show that people are faster to report the names of people like them, compared to people who mismatch them in terms of gender or race.

As the Reviewer notes above, our original tasks included a name learning component, in which participants had to study the names of eight people and were then tested on those names until they reached ceiling accuracy. This test occurred both at the start of the study and then repeatedly afterwards, ensuring that participants did not forget the names. From this data, we can analyse the speed with which participants were able to report these names. We did this in two ways. First, we conducted an exploratory analysis on our Chinese data (Experiment 4), finding that, for trials where participants provided an accurate response, they were fastest to report the names of people who matched them on gender and race. We then pre-registered a matched analysis for the data from our three English studies.

The results were clear cut. Participants were fastest to select names for people who matched them on both gender and race. On average, they were about 80ms slower to select names for people who matched them on one demographic dimension but did not match on the other dimension. They were then a further 50ms slower to name people who did not match them on either gender or on race. Thus, participants were faster to remember the names of people who were more like them, indicating that those names are indeed more accessible. This accessibility advantage naturally explains the Like Me grammatical ordering effects that we saw in our main studies.

We hope that this convinces the Reviewer of our accessibility account. We additionally complement this new analysis with further evaluations of the self-serving bias account, showing that while communicated agency and communicated speaker identity vary across grammatical constructions, the Like Me bias does not. Thus, we believe that there is strong evidence that the Like Me bias is caused by accessibility and incrementality.

(Note that these new analyses are now reported after Experiment 3. For brevity, we do not repeat the significant new text here.)

In relation to Nature Human behaviors guidelines on significant advance I cannot see that this paper presents such novelty to be published in this journal. I really do appreciate the paper and find that it provides some conceptual novelty in how individuals from different groups are presented, but more research is needed that provides stronger evidence for the “like-me” hypothesis.

Thus, it is the novelty in relation to past literature on self-serving biases in combination how well the different studies can disentangle social effects from cognitive effects that brings me to a conclusion to not endorse publication of the present paper.

We appreciate the Reviewer’s comments but of course disagree that our finding has limited novelty. First, as discussed above, we believe that we do now have the strong evidence to disentangle social and cognitive effects here. Second, to our knowledge, there is no prior evidence that factors like self-serving biases affect word order in the manner tested here (although we would be happy to be corrected). So for these reasons we believe that our work truly provides a novel bridge between cognitive psychology, sociolinguistics, and the social psychology of languages.

Decision Letter, first revision:

** Please ensure you delete the link to your author homepage in this e-mail if you wish to forward it to your co-authors. **

Our ref: NATHUMBEHAV-23082568A

13th February 2024

Dear Dr. Brough,

Thank you for your patience as we’ve prepared the guidelines for final submission of your Nature

Human Behaviour manuscript, "Accessibility and incrementality cause "Like Me" race and gender biases in human language production" (NATHUMBEHAV-23082568A). Please carefully follow the step-by-step instructions provided in the attached file, and add a response in each row of the table to indicate the changes that you have made. Please also address the additional marked-up edits we have proposed within the reporting summary. Ensuring that each point is addressed will help to ensure that your revised manuscript can be swiftly handed over to our production team.

We would hope to receive your revised paper, with all of the requested files and forms within two-three weeks. Please get in contact with us if you anticipate delays.

Nature Human Behaviour offers a Transparent Peer Review option for new original research manuscripts submitted after December 1st, 2019. As part of this initiative, we encourage our authors to support increased transparency into the peer review process by agreeing to have the reviewer comments, author rebuttal letters, and editorial decision letters published as a Supplementary item. When you submit your final files please clearly state in your cover letter whether or not you would like to participate in this initiative. Please note that failure to state your preference will result in delays in accepting your manuscript for publication.

In recognition of the time and expertise our reviewers provide to Nature Human Behaviour's editorial process, we would like to formally acknowledge their contribution to the external peer review of your manuscript entitled "Accessibility and incrementality cause "Like Me" race and gender biases in human language production". For those reviewers who give their assent, we will be publishing their names alongside the published article.

Cover suggestions

We welcome submissions of artwork for consideration for our cover. For more information, please see our guide for cover artwork.

ORCID

Non-corresponding authors do not have to link their ORCIDs but are encouraged to do so. Please note that it will not be possible to add/modify ORCIDs at proof. Thus, please let your co-authors know that if they wish to have their ORCID added to the paper they must follow the procedure described in the following link prior to acceptance: <https://www.springernature.com/gp/researchers/orcid/orcid-for-nature-research>

Nature Human Behaviour has now transitioned to a unified Rights Collection system which will allow our Author Services team to quickly and easily collect the rights and permissions required to publish

your work. Approximately 10 days after your paper is formally accepted, you will receive an email in providing you with a link to complete the grant of rights. If your paper is eligible for Open Access, our Author Services team will also be in touch regarding any additional information that may be required to arrange payment for your article.

Please note that *Nature Human Behaviour* is a Transformative Journal (TJ). Authors may publish their research with us through the traditional subscription access route or make their paper immediately open access through payment of an article-processing charge (APC). Authors will not be required to make a final decision about access to their article until it has been accepted. Find out more about Transformative Journals

[REDACTED]

Best regards,
[REDACTED]

On behalf of
[REDACTED]

Reviewer #1:

Remarks to the Author:

I've read the revised manuscript and responses to all of the reviewers, and feel the authors have addressed the concerns raised. The revision is even stronger, more clear and more interesting. The new data included is compelling and I would be delighted to see the paper appear in NHB. -Adele Goldberg

Reviewer #3:

Remarks to the Author:

I am satisfied with the authors' responses to my comments and questions in the previous round. The authors have clarified all the questions I had about the design and analysis of Experiment 4, and made necessary revisions in the manuscript to correct the inconsistencies. I also appreciate the additional work that the authors have included in the revised manuscript, which strengthens the theoretical impact of the paper.

I have no further questions about the paper. I do have one minor, side comment about the Chinese

stimuli. In the authors' response to one of my questions (top of p12 in the rebuttal letter), I couldn't help but notice that the example in the rightmost box is "[名字]正在对话[名字]". It is a very unnatural- and borderline ungrammatical-sentence in Chinese, as the verb 对话 ("v. to talk") cannot be used as a transitive verb. One would have to say "与[名字]对话" (literally "with someone talk") to mean "talk with someone". I see from the manuscript that this verb 对话 was actually not used in Experiment 4, so there's nothing to worry about for the current paper; but in future research, I'll recommend that the authors do a more rigorous check of the goodness of all the Chinese stimuli.

Signed
Yao Yao

Reviewer #4:

Remarks to the Author:

The authors have adeptly addressed critical concepts within the manuscript and the added studies enhances the strengths of their paper itself. The paper provides a coherent and comprehensive story, and I appreciate the manuscript in its present form. I recommend it for publication and have only two minor comments.

First, I have concerns regarding p-values as "marginally significant" when p-values exceed 0.1. This type of reporting can be misleading, particularly in light of the observed insignificance in the overall test pertaining to the "like me" bias among black women. A reevaluation of the statistical significance threshold is recommended to ensure the accuracy and rigor of the reported findings.

Second, given the intricate nature of biases such as the "like me" effect, I would also suggest that the authors discuss potential influencing factors in relation to the findings. For example, do the authors come up some more insights into why gender emerges as a more robust indicator of the "like me" bias. Another possible avenue for further research would be to explore age as a significant predictor of perceived similarity among individuals of the same age. Thus, are there potential circumstances under which the "like me" effect might be attenuated or weakened.

Final Decision Letter:

Dear Dr Brough,

We are pleased to inform you that your Article "Cognitive causes of Like Me race and gender biases in human language production", has now been accepted for publication in *Nature Human Behaviour*.

Please note that *Nature Human Behaviour* is a Transformative Journal (TJ). Authors may publish their research with us through the traditional subscription access route or make their paper immediately open access through payment of an article-processing charge (APC). Authors will not be required to make a final decision about access to their article until it has been accepted. Find out more about Transformative Journals

Authors may need to take specific actions to achieve compliance with funder and institutional open access mandates. If your research is supported by a funder that requires immediate open access (e.g. according to Plan S principles) then you should select the gold OA route, and we will direct you to the compliant route where possible. For authors selecting the subscription publication route, the journal's

standard licensing terms will need to be accepted, including self-archiving policies. Those licensing terms will supersede any other terms that the author or any third party may assert apply to any version of the manuscript.

In approximately 10 business days you will receive an email with a link to choose the appropriate

publishing options for your paper and our Author Services team will be in touch regarding any additional information that may be required.

With best regards,

[REDACTED]